# Role of *Phyllanthus niruri* on the modulation of stress and immune responses in Nile tilapia, *Oreochromis niloticus*

Md Ibrahim¤ᵃ, Mursalin Khan¤ᵇ◉, Israt Mishu◉, Ishrat Jahan◉, Ahmed Mustafa◉*

Department of Biological Sciences, Purdue University Fort Wayne, Fort Wayne, Indiana, United States of America

◉ These authors contributed equally to this work.
¤a Current address: Charles River Laboratories, Ashland, Ohio, United States of America
¤b Current address: Department of Biological Sciences, Auburn University, Auburn, Alabama, United States of America
* mustafaa@pfw.edu

**Data Availability Statement:** All relevant data are within the manuscript and its Supporting information files.

## Abstract

The use of nutraceuticals in aquaculture is getting attention to minimize oxidative stress and increase immunity of aquatic animals. In our experiment, we tested the potency of *Phyllanthus niruri*, as a stress-reducing and immune-stimulating agent in Nile tilapia, *Oreochromis niloticus*. We reared fish in a recirculating system for 90 days at low (5 g/L) and high (30 g/L) densities. We fed half of the low and high-density fish with commercial feed (control feed) and the other half, with 5% *Phyllanthus niruri* incorporated into commercial feed (supplemented feed). We assessed plasma cortisol, blood glucose, packed cell volume, plasma proteins, phagocytic capacity, respiratory burst activity, and lysozyme activity. None of these parameters showed any significant difference among the experimental groups. Based on our findings, we conclude that *Phyllanthus niruri* does not have any potential role in modulating stress and immune response in fish as a nutraceutical.

## 1. Introduction

The extensive production of aquatic products from the aquaculture sector has made it one of the fastest growing food generating industries in the world [1]. But, like other industries, aquaculture also has its own problems. Intensification of culture methods such as overcrowding, handling, fluctuating water quality such as temperature, salinity, oxygen etc., and inadequate nutritional causes disease outbreak in culture systems that may lead to total or partial loss of production [2–4]. High density culture systems and poor hygienic conditions also help spread of pathogens causing a high mortality in fish [5–7].

Stress is a physiological reaction of an organism in response to a stimulus that causes homeostatic imbalance [8]. As a primary stress response, when fish is exposed to a stressor, its central nervous system (CNS) recognizes the stressor as a threat, and helps release catecholamines by chromaffin cells, located in the head kidney [9]. Cortisol is released after the secretion of catecholamines, upon the activation of hypothalamus-pituitary-interrenal axis (HPI)

**Funding:** The author(s) received no specific funding for this work.

**Competing interests:** The authors have declared that no competing interests exist.

[9]. Release of cortisol promotes glycogenolysis and glucogenesis in the liver [10]. Release of cortisol is slower than catecholamines, but it has a lag time of several minutes, which makes it a prominent indicator of stress in fish. Generally, increase in the plasma cortisol level stimulate to decline in the circulating lymphocytes which leads to lower survivability [8]. Secondary stress response which is followed by primary stress response, is mainly related to blood and tissue alterations [11]. Changes occur in metabolic (e.g. plasma glucose), hematological (e.g. hematocrit), hydromineral (e.g. plasma proteins), and tissue structure level [8]. For example, fish increase the level of blood glucose and packed cell volume and decrease the blood clotting time when exposed to stress. Tertiary stress response is considered a whole-body response, or a response at population or community level [8]. It has been well established that stress makes fish susceptible to diseases [12]. During stress responses, fish allocate their metabolic energy to regain the homeostatic status, which in turn affects the growth, longevity, and reproductive capacity of fish [13]. Reduced and negative growth are commonly observed in fish when exposed to stress; therefore, changes in biomass, condition factor, and food conversion efficiency are reliable indicators of stress response in fish [14].

To combat with the stress and its consequences, farmers use various kinds of drugs and chemicals in culture systems to prevent infections and treat diseases. The use of drugs may be helpful in short term, but in long term, this is hazardous for fish, consumers, and for the environment [15]. Extensive use of antibiotics may produce antibiotic resistance in fish [16, 17], and residual effect of antibiotics in fish muscle may harm human consumers [5, 18]. One of the ways to deal with infection and diseases in aquaculture is the development and use of vaccines, which mainly elicit specific immune response. Although vaccination could be used as a prophylactic measure to control diseases, it is expensive, pathogen specific, but it requires to prepare vaccine long time [19–23]. As vaccines are pathogen specific and only applicable for specific or adaptive immune response, they are not effective to elicit innate immune response. Innate or non-specific immune response is the first line defense mechanism in body against invading organism, and it comes into action immediately or within short time of antigen's appearance in the body. The components of innate or non-specific immune system are anatomical barriers, secretory molecules, macrophages, monocytes, granulocytes, humoral elements, lysozyme, and complement system [24–27]. In aquatic species, mostly in fish and shellfish, innate immune system consists of activation of neutrophil, production of reactive oxygen species, and activation of inflammatory factors [28, 29]. Phagocytosis is the main mediator in the non-specific immune system to engulf, neutralize and kill virus, bacteria, and parasites [30].

Due to the limitations of using vaccines in culture medium, it is a demand of time in aquaculture to look for alternative measure that will enhance and stimulate the immune system and prevent diseases as well. Use of nutraceuticals is a promising alternative to serve this purpose. Nutraceuticals are any food or food components that have both nutritional value and therapeutic value. Nutraceuticals can be obtained from both animals and plants, but plant-originated nutraceuticals are widely available and used in aquaculture practice [19, 20]. Due to wide range of availability, cost effectiveness, and sustainability, nutraceuticals have been a popular alternative to chemotherapeutics to strengthen the defense mechanism in fish, and hence, disease prevention [19, 20].

In this study, we have selected *Phyllanthus niruri* (PN) as a nutraceutical to investigate its effect on overall wellbeing of fish. PN also known as stone breaker, an herbal medicinal plant under Euphorbiaceae family, is available in tropical areas and Amazon Forest [31]. The other names of this plant are *Chanca piedra*, bhoomi amalaki, bhui-amla etc. [31]. The genus *Phyllanthus* contains nearly 600 species of shrubs [31]. Several research suggested the presence of biologically active components lignans, glycosides, flavonoids, alkaloids, ellagitannins,

phenolics, steroids, and essential oils in PN [31]. Studies in fish and other species found it effective in antioxidant defense [32], antiviral activity [33], improving immune response [34], increasing growth [35], reducing liver toxicity [36–39], reducing blood sugar [40], and relieving pain [41]. To the best of our knowledge, no research has been done to investigate the effects of PN in the physiological and immunological responses of Nile tilapia, *Oreochromis niloticus*, however some other fish of the same and mentioned in few studies [42, 43]. Also, due to its role as antidiabetic (reduction of glucose), we hypothesized that it would work as stress relieving agent as increase of blood glucose is an indication of stress event in fish. We selected Nile tilapia, *Oreochromis niloticus*, because it is a widely cultured species in aquaculture and has the second position after carps, in terms of farmed fish production [44].

The overall goal of our research was to promote an ecofriendly aquaculture by using plant origin nutraceutical as feed additives. We hypothesized addition of PN as a feed additive would reduce the effects of stress in fish, and also would stimulate the immune system because of the presence of its active ingredients that has been shown beneficial to humans. The specific aim of our research is to observe the effects of PN-supplemented diet on the modulation of density stress and immune responses in Nile tilapia, reared in inland recirculating system. To determine stress and immune responses, we investigated physiological parameters such as plasma cortisol, blood glucose, packed cell volume, plasma protein, and condition factor, and immunological parameters such as phagocytic capacity, macrophage respiratory burst activity, and lysozyme activity.

## 2. Materials and methods

### 2.1 Fish acquisition and maintenance

Nile tilapia, *Oreochromis niloticus*, fingerlings with an average length of 12.00±0.218 cm and an average weight of 33.67±0.427 g were obtained from a certified tilapia farm (Troyer Aqua, IN). Upon arrival, fish were acclimated to the lab environment for two weeks. During this acclimation period, fish were fed Purina Aquamax Fingerling Starter 300 (PMI Nutrition International, MO, USA) at 2% of body weight once per day. Fish were given a natural photoperiod of 12:12 hours dark-light cycle throughout the whole study, and 3–5% water was exchanged on daily basis. Water quality pH, dissolved oxygen, ammonia, and temperature levels) was monitored regularly, and fish were taken care of following the guidelines of Purdue University Fort Wayne and an approved animal care protocol of PACUC (Purdue Animal Care and Use Committee).

### 2.2 Experimental design

For a 90 day long chronic study, fish were reared in inland recirculating system at low density (5 g/L) and high density (30 g/L). There were four experimental groups in duplicate in each system: 1) Low density fed control feed (LDCF), 2) Low density fed PN-supplemented feed (LDSF), 3) High density fed control feed (HDCF), and 4) High density fed PN-supplemented feed (HDSF).

### 2.3 Feed preparation

Aquamax Fingerling Starter 300 was used as control feed, and PN powder (Raintree Formulas, IA, USA) was added (by coating on the pellet) with control feed at 5% concentration to prepare supplemented feed. The 5% concentration was used for main study because it worked better than other concentrations in our pilot study [45]. Fish were fed once per day at 2% of total fish biomass.

## 2.4 Fish sampling

Six fish per group (three from each replicate) were sampled at week 0, 4, 8, and 12. On each sampling day fish were randomly picked by a net and rapidly euthanized using tricaine methane sulfonate (MS-222) (Sigma-Aldrich, MO, USA) at 200 mg/L, as high dose of MS-222 causes rapid immobilization of fish without changing the cortisol level [46]. Following euthanization, length and weight of the sampled fish were measured. Blood was then collected from the caudal vein of fish. After drawing the blood, fish were sacrificed for collecting macrophages from the head kidney for measuring macrophage phagocytic capacity and respiratory burst activity.

## 2.5 Analyses of samples

To investigate the effect of PN on Nile tilapia, physiological and immunological responses were measured. Physiological parameters such as plasma cortisol, blood glucose, packed cell volume, plasma proteins, and condition factor were determined from sampled fish. Immunological parameters such as phagocytic capacity, lysozyme activity, and respiratory burst activity were measured from sampled fish.

For plasma cortisol, blood was centrifuged at 1000 rpm for 5 min to separate plasma from the blood. Plasma samples were then kept in microcentrifuge tubes and preserved at -80˚C. Plasma cortisol of all samples was determined, later, using a Cortisol ELISA kit (ENZO Life Sciences, NY, USA). Blood glucose was measured using Abbott Free-Style Lite Glucometer (Abott Diabetes Care, CA, USA). Use of standardized glucometer to measure the blood glucose in fish was validated by Iwama et al. 1995 [47]. After measuring the blood glucose, a portion of remaining blood, was taken in a capillary tube and then centrifuged in a micro-centrifuge at 10,000 rpm for 5 mins. After centrifugation, packed cell volume was measured using micro-hematocrit capillary tube reader (Monoject Scientific, MO, USA). The total plasma proteins of blood plasma were measured by a refractometer (VEEGEE Scientific Inc., WA, USA).

For macrophage phagocytic capacity, macrophages were isolated, and their phagocytic capacity was determined following Brown et al. 1996 [48] and modified by Mustafa et al. 2000 [49].

This assay determined the proportion of macrophages capable of phagocytizing formalin killed bacteria, *Bacillus megaterium*. At least 100 cells were counted to determine the phagocytic capacity. A cell was considered as a positive if it engulfed at least 5 bacteria in its cytoplasm. Then a proportion of positive cells was calculated among 100 cells [49]. The respiratory burst activity of phagocytic macrophage cells was measured by the reduction of nitro-blue tetrazolium (NBT) by intracellular superoxide radicals produced by leukocytes stimulated with phorbol myristate acetate (PMA). Macrophage respiratory burst and phagocytic activity were measured following the methods described by Brown et al. 1996 [48] and Secombes 1990 [50]. During this reaction NBT is reduced by $O_2$ into an insoluble blue formazan.

The lysozyme activity in fish blood plasma was assessed using a modified microplate lysozyme turbidity assay as described by Ellis (1990) [51]. Initially, blood samples were centrifuged at 1000 rpm for 5 mins to separate the plasma. Subsequently, two separate solutions were prepared by dissolving 0.14 g of Na2HPO4 (Sigma-Aldrich, St. Louis, MO) and NaH2PO4 (Sigma-Aldrich, St. Louis, MO) each in 20 mL of deionized water. A 0.05 M sodium phosphate buffer was then created by mixing 16.3 mL of the NaH2PO4 solution with 3.7 mL of the Na2HPO4 solution. For the assay, 4.925 mL (0.05 M sodium phosphate buffer) of this buffer was combined with 0.985 mg of *Micrococcus luteus* (previously *Micrococcus leisodeikticus*) powder to form the working bacterial buffer. The assay involved adding 50 µL of plasma and 50 µL of the bacterial buffer to each well of a 96-well plate, with all samples run in triplicate.

Absorbance measurements were taken at 570 nm immediately after setup and at 0, 1, 5, 10, and 15-minute intervals. A blank consisting of phosphate buffer and bacteria solution was used for calibration, and its absorbance was subtracted from that of the samples. Lysozyme activity was quantified based on the reduction in absorbance from 0 to 5 minutes, with a greater decrease indicating higher enzymatic activity.

Condition factor is the ratio of fish weight to the cubic length of fish which measures the overall wellbeing and nutritional status of the fish. Condition factor is denoted by K and mostly known as Fulton's condition factor [52]. Condition factor $(K) = \frac{W}{L^3} \times 100$, where W = weight of fish in gram, L = total length of fish in cm.

## 2.6 Statistical analysis

All data were analyzed by Sigmaplot 14.0. A one-way analysis of variance (ANOVA) was used to compare the means of all data and for further analysis through the Tukey's HSD (Honestly Significant difference) at level of significance <0.05 to compare the difference among the experimental groups. Data were represented as Mean ± SEM (Standard Error of Mean).

## 3. Results

Over time, plasma cortisol levels diminished across all experimental groups. By week 12, the LDCF group exhibited a significant reduction (P<0.05) in cortisol levels compared to all experimental groups at weeks 0 and 4, as well as compared to both the LDCF andHDCF groups at week 8. By the same time point, all other groups also showed significant reductions in cortisol levels relative to their respective measurements at week 0 (refer to S1A Fig). However, when considering all groups collectively, no significant differences in cortisol levels were observed among the experimental groups (Fig 1A).

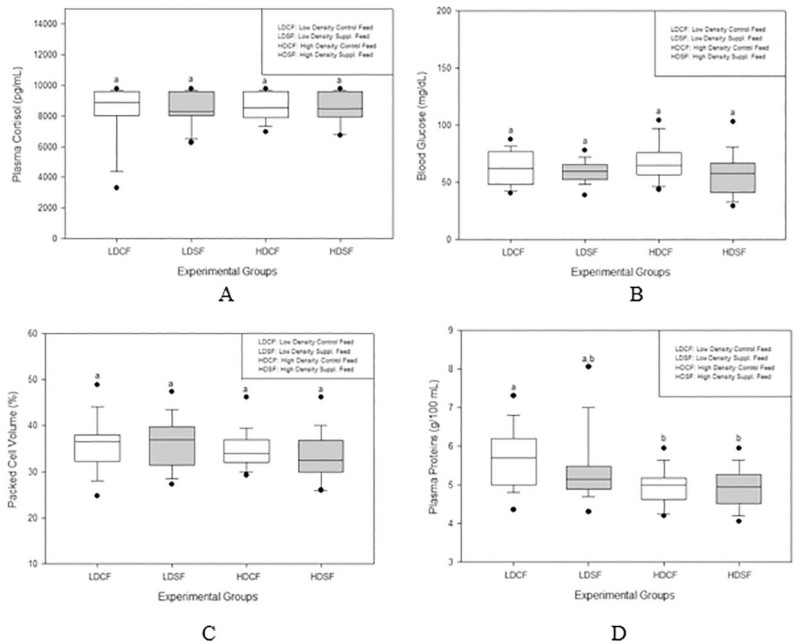

**Fig 1. A) Plasma cortisol, B) Blood glucose, C) Packed Cell Volume, and D) Plasma protein in different experimental groups over the three months experimental period.** Different letters indicate significant difference among different groups (One-way ANOVA, Tukey test, P<0.05). Each boxplot indicates the median (midline in the box) and the 5th, 25th, and 75th percentiles from bottom to top. Filled circles are far from outliers.

In both low- and high-density conditions, groups receiving supplemented diets exhibited lower glucose levels compared to those on control feeds at weeks 4, 8, and 12; however, these differences did not reach statistical significance (S1B Fig). A comprehensive analysis of all sampling points indicated that glucose levels were consistently lower in the supplemented diet groups compared to the control feed groups (Fig 1B).

Throughout the sampling periods, no significant differences were observed in packed cell volume (PCV) percentages among the experimental groups. Additionally, the PCV levels did not display any discernible trends over the course of the study (S1C Fig). An aggregated analysis of all sampling points also revealed no significant variations in PCV among the groups (Fig 1C).

During the sampling stages at weeks 4, 8, and 12, the LDCF and LDSF groups exhibited higher plasma protein levels compared to the supplemented diet groups. Specifically, the LDCF group demonstrated a significant increase (P<0.05) in plasma protein levels at weeks 4 and 12 compared to its baseline levels at week 0 (S1D Fig). When analyzing data across all sampling stages, the LDCF group showed significantly higher (P<0.05) plasma protein levels than both the HDCF and HDSF groups (Fig 1D).

Throughout the experimental period, an increase in phagocytic capacity was observed across all experimental groups. The groups fed supplemented diets exhibited higher phagocytic capacities compared to those on control feeds at weeks 4, 8, and 12 time points, although these increases did not achieve statistical significance (S2A Fig). A comprehensive analysis incorporating all groups revealed no significant differences in phagocytic capacity among the groups; however, the supplemented diet groups consistently demonstrated higher levels of phagocytic capacity than the control feed groups (Fig 2A).

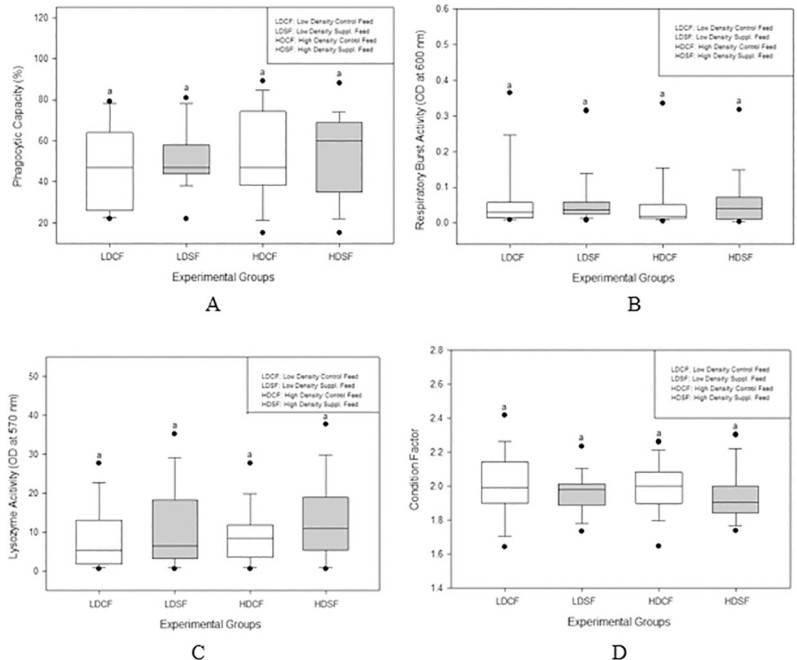

**Fig 2. A) Macrophage phagocytic capacity, B) Macrophage respiratory burst activity, C) Serum lysozyme activity, and D) Fish condition factor in different experimental groups over the three months experimental period.** Different letters indicate significant difference among different groups (One-way ANOVA, Tukey test, P<0.05). Each box plot indicates the median (midline in the box) and the 5th, 25th, and 75th percentiles from bottom to top. Filled circles are far from outliers.

Throughout the sampling periods, no significant differences in respiratory burst activity were observed among the experimental groups (S2B Fig). Similarly, when data from all sampling stages were aggregated and analyzed collectively, no significant differences were detected in respiratory burst activity across the groups (Fig 2B).

At week 4, the supplemented diet groups under low density conditions and at week 12 for high density conditions demonstrated elevated levels of lysozyme activity compared to the control feed groups (S2C Fig). When analyzing data from all combined sampling stages, the supplemented diet groups consistently showed higher lysozyme activity than the control feed groups (Fig 2C). However, lysozyme activity measurements for the week 8 sampling period were not obtained due to a malfunction in the 96-well plate reader.

Throughout the experimental period, all experimental groups exhibited condition factors greater than 1.5, indicating that the fish maintained good health. Analysis of the data revealed no significant differences in condition factors among the experimental groups at any individual sampling stage (S2D Fig). Additionally, when data from all four sampling stages were aggregated and analyzed collectively, no significant differences in condition factors were observed across the groups (Fig 2D).

## 4. Discussion

A detailed examination of various physiological and immunological parameters was conducted to evaluate the impact of *Phyllanthus niruri* on the overall health and stress reduction in Nile tilapia. The study noted a decrease in blood glucose levels and an enhancement in lysozyme activity associated with *P.niruri* supplementation, suggesting beneficial effects. However, a comprehensive analysis of all measured parameters did not conclusively demonstrate that *P. niruri* serves as an effective stress reliever in this context. Typically, parameters such as plasma cortisol, blood glucose, packed cell volume (PCV), and plasma proteins are known to increase under stress conditions in fish, with reductions indicating potential stress relief [8, 49, 53]. Conversely, parameters like respiratory burst activity, lysozyme activity, phagocytic activity, and condition factor typically decrease under stress, with increases suggesting an improvement in stress resilience [49, 54].

Plasma cortisol concentrations did not vary significantly among the experimental groups, suggesting that Nile tilapia may adapt readily to crowding stress. The experimental setup included a control density of 5 g/L and a high density of 30 g/L, which was hypothesized to induce stress. However, the high density may not have been sufficient to elicit a stress response in tilapia, a species known for its robustness and ability to tolerate a wide range of environmental conditions. Tilapias are remarkably resilient, capable of rapid growth and survival across extreme temperature ranges, from as low as 10°C to as high as 40°C [55, 56]. They can function under oxygen saturation levels below 20%, although they revert to anaerobic metabolism and begin gulping ambient air under such conditions [57].

Cortisol is a commonly used biomarker for stress in fish, typically upregulated in acute stress scenarios and rapidly secreted, but normalizing or dissipating after several hours of stress exposure [8, 58]. In many fish species, cortisol levels peak within 1 hour of stress onset and return to baseline within 6 hours [59]. For example, in red drum subjected to handling stress, cortisol levels spike but return to basal levels within 48 hours [60]. In cases of chronic stress, cortisol levels may decline due to either adaptation of the endocrine system to the stressor or exhaustion from prolonged hyperactivity [61, 62]. During chronic stress, interrenal cells may also become less responsive to adrenocorticotropic hormone (ACTH) or other pituitary hormones, leading to reduced cortisol production.

In aquaculture, the evaluation of stress responses is important, with plasma cortisol and blood glucose levels serving as key biomarkers. Typically, an increase in cortisol secretion, triggered by stress, is accompanied by a rise in blood glucose concentrations, facilitating an immediate energy supply to cope with the stressor. However, in our study, glucose concentrations were observed to decrease in high-density groups supplemented with PN extract at weeks 8 and 12 [63, 64]. This reduction is consistent with the known properties of PN, a plant utilized extensively across South America and Southeast Asia for its medicinal benefits [31].

PN has a broad spectrum of therapeutic applications, including the treatment of diabetes, inflammation, and various biliary and urinary conditions. The plant is revered for its anti-inflammatory, antiviral, and glucose-lowering properties, derived from all its parts—leaves, stems, and roots [31]. Specifically, PN has demonstrated significant antidiabetic activity, as evidenced by its ability to reduce blood glucose levels in diabetic rats through antioxidant activities that mitigate oxidative stress [40, 65]. While the typical physiological response to stress involves a concomitant elevation of both cortisol and glucose levels, discrepancies between these two markers are not uncommon. Such anomalies have been documented in various studies, where glucose levels have increased independently of cortisol [66]. This divergence might be attributed to alternative biochemical pathways influencing glucose metabolism, such as the action of catecholamines, which can also elevate glucose levels in the blood, especially during acute stress phases [66, 67]. Over time, chronic stress may lead to a normalization of glucose levels, indicating an adaptive metabolic response [68].

In this investigation, the packed cell volume (PCV) values observed across all experimental groups were within the established normal range for Nile tilapia, which is 22–45% [69]. To date, there have been no studies evaluating PCV in Nile tilapia or any other fish species following a diet supplemented with PN. A similar observation was noted for plasma protein concentrations, which also remained within the normal expected range for Nile tilapia, documented as 2.9–6.6 g/100 mL [70]. Consequently, based on the measurements of these two physiological parameters PCV and plasma proteins, it was not possible to ascertain any stress-relieving effects of PN supplementation in this study.

In this study, no consistent trends were observed in immunological responses, including phagocytic capacity, respiratory burst activity, and lysozyme activity, across the experimental groups. The lack of previous scientific research on the use of PN as a dietary supplement for Nile tilapia or other fish species precluded the possibility of comparing these immunological findings with those of other studies.

Additionally, the condition factor, which is commonly used as an indicator of the overall health of fish, was maintained above 1 in all experimental groups. This metric suggests that the fish were in good health throughout the various treatment groups. In tilapia, a condition factor greater than 1 is generally indicative of favorable health conditions [71].

## 5. Conclusion

This research investigated physiological and immunological responses of Nile tilapia, fed with *P. niruri*-supplemented diet to determine whether it reduces the stress and stimulates the immune system, as this plant has shown multiple health benefits in human, particularly in reduction blood sugar level. No clear stress indication was found in high density groups based on the findings of plasma cortisol and blood glucose concentrations. Although not significant, blood glucose concentrations were higher in high density control feed groups than in the supplemented diet groups across all sampling periods. Except for plasma proteins, other physiological parameters did not vary significantly among all sampling stages. By analyzing the

immunological parameters, no evidence was found to suggest that this supplemented diet positively modulated the immune response in Nile tilapia.

## Supporting information

**S1 Fig. A) Plasma cortisol, B) Blood glucose, C) Packed Cell Volume, and D) Plasma proteins in different experimental groups at week 0, 4, 8, and 12 sampling period over the 90 day experiment.** Data presented as means ± SEM, and N = 6 fish/group. Different letters and numbers indicate significant difference among different groups (One-way ANOVA, Tukey test, P<0.05).
(TIF)

**S2 Fig. A) Macrophage phagocytic capacity, B) Macrophage respiratory burst activity, C) Serum lysozyme activity, and D) Fish condition factor in different experimental groups at week 0, 4, 8, and 12 sampling period over the 90 day experiment.** Data presented as means ± SEM, and N = 6 fish/group. Different letters indicate significant difference among different groups (One-way ANOVA, Tukey test, P<0.05).
(TIF)

**S1 Dataset.**
(XLSX)

## Acknowledgments

We would like to thank the Department of Biological Sciences at Purdue University Fort Wayne for providing us opportunity to utilize its facilities to conduct this experiment.

## Author Contributions

**Conceptualization:** Md Ibrahim, Ahmed Mustafa.

**Data curation:** Md Ibrahim.

**Formal analysis:** Md Ibrahim.

**Investigation:** Md Ibrahim, Mursalin Khan.

**Methodology:** Md Ibrahim, Mursalin Khan, Ahmed Mustafa.

**Project administration:** Ahmed Mustafa.

**Resources:** Ahmed Mustafa.

**Supervision:** Ahmed Mustafa.

**Validation:** Ahmed Mustafa.

**Writing – original draft:** Md Ibrahim.

**Writing – review & editing:** Mursalin Khan, Israt Mishu, Ishrat Jahan, Ahmed Mustafa.

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
