## [Decision Letter · Decision Letter 0]

24 Jun 2024

PONE-D-24-21421Role of Phyllanthus niruri on the modulation of stress and immune responses in Nile tilapia, Oreochromis niloticusPLOS ONE

Dear Dr.  Mustafa,

Thank you for submitting your manuscript to PLOS ONE. After careful consideration, we feel that it has merit but does not fully meet PLOS ONE’s publication criteria as it currently stands. Therefore, we invite you to submit a revised version of the manuscript that addresses the points raised during the review process.

We look forward to receiving your revised manuscript.

Kind regards,

Ishtiyaq Ahmad, Ph.D

Academic Editor

PLOS ONE

Reviewers' comments:

Reviewer's Responses to Questions

**Comments to the Author**

1. Is the manuscript technically sound, and do the data support the conclusions?

Reviewer #1: Yes

2. Has the statistical analysis been performed appropriately and rigorously? 

Reviewer #1: Yes

3. Have the authors made all data underlying the findings in their manuscript fully available?

Reviewer #1: Yes

4. Is the manuscript presented in an intelligible fashion and written in standard English?

Reviewer #1: Yes

5. Review Comments to the Author

Reviewer #1: Manuscript Number: PONE-D-24-21421

Title: Role of Phyllanthus niruri on the modulation of stress and immune responses in Nile tilapia, Oreochromis niloticus

Journal: PLOS ONE

This current study examined Phyllanthus niruri supplementation diet in Nile tilapia, Oreochromis niloticus to improved stress and immune responses as useful one of important medicinal herb additive aquatic animal health management. This Ms was found to clear some major issue before consider of the Ms in this journal.

Line 22: ‘reduce stress’ change to ‘minimize oxidative stress’

Line 24: ‘for Nile tilapia’ change to ‘in Nile tilapia, Oreochromis niloticus’

Line 35: include ‘generating’ after food

Line 40: Include ‘https://doi.org/10.1080/10641260500320845’

Line 48, 61, 63, 64: ‘induces reduction’ to change to ‘stimulate to decline’

Line 67: ‘require a long time to develop a vaccine’ change to ‘but it require to prepare vaccine long time’

Line 68: include ‘https://doi.org/10.1016/j.fsi.2010.02.026; https://doi.org/10.1016/j.aquaculture.2010.09.011’

Line 71: ‘within the hours’ change to ‘within short time’

Line 74, 76: include ‘ https://doi.org/10.1016/j.aquaculture.2011.03.039; https://doi.org/10.1016/j.aquaculture.2011.07.022’

Line 79: ‘options’ change to ‘measure’

Line 80: ‘option’ change to ‘alternative’

Line 83: include ‘and used in aquaculture practice’ after available

Line 95: no research has been done to investigate the effects of PN in the physiological and immunological responses of Nile tilapia, Oreochromis niloticus, but some other fish of the same and mentioned in this Ms:

https://www.ncbi.nlm.nih.gov/pmc/articles/PMC5090155/pdf/ijvr-17-200.pdf

https://docs.lib.purdue.edu/dissertations/AAI10182448/

https://sdiopr.s3.ap-south-1.amazonaws.com/doc/Rev_AJFAR_77263_Jor_A.pdf

https://www.fisheriesjournal.com/archives/2017/vol5issue1/PartE/4-6-63-494.pdf

https://jchr.org/index.php/JCHR/article/view/1721

https://www.fisheriesjournal.com/archives/2023/vol11issue1/PartA/11-1-8-767.pdf

Line 80: Here provide Water quality PH , dissolved oxygen, ammonia, and temperature levels..

Line 152: ‘1000 rpms for 5 mins’ change to ‘1000 rpm for 5 min’

Line 159, 176: ‘minutes’ change to ‘min’

Line 201: ‘Low-Density Control Feed (LDCF)’ change to ‘LDCF’

Line 204: ‘High-Density Control Feed (HDCF)’ change to ‘HDCF’

Line 217: ‘Low-Density Control Feed (LDCF) and Low-Density Supplemented Feed (LDSF)’ change to ‘LDCF and LDSF’

Line 219: Include ‘PN” before supplemented

Line 222: ‘the High-Density Control Feed (HDCF) and High-Density Supplemented Feed (HDSF)’ change to ‘HDCF and HDSF’

Line 259: ‘different’ change to ‘the’

Line 281, 301, 305, 308: ‘Phyllanthus niruri (P. niruri)’ change to ‘PN’

Line 282, 284, 287, 318: ‘P. niruri’ change to ‘PN’

Line 310: Related citation include

6. PLOS authors have the option to publish the peer review history of their article (what does this mean?). If published, this will include your full peer review and any attached files.

Reviewer #1: No

---

## [Author Response · Author response to Decision Letter 0]

6 Aug 2024

We would like to thank the reviewer for the valuable comments/suggestions. 

We have incorporated all the corrections as suggested. (Please see the revised manuscript with track change).

Please find attached the revised manuscript titled, “Role of Phyllanthus niruri on the modulation of stress and immune responses in Nile tilapia, Oreochromis niloticus”- for publication in PLOS ONE.

We have revised the manuscript and incorporated information as suggested by the reviewers. We have uploaded the manuscript in two forms: Manuscript and Manuscript with track change.

We have selected PLOS ONE because of its multi-disciplinary and inter-disciplinary nature of publishing advance science for the benefits of the society and the nature. Our research falls into these categories and will disseminate our findings for the betterment of animal health and for saving the nature. In recently published research, researchers have shown that marine nutraceuticals are sources of alternative therapeutics. In our work, we have emphasized on the use similar idea, that is, investigating the properties of nutraceuticals, Phyllanthus niruri. Scientists involved in research on aquatic animals - will be mostly benefitted from this article.

We will appreciate your timely review and acceptance.

---

## [Decision Letter · Decision Letter 1]

20 Aug 2024

Role of Phyllanthus niruri on the modulation of stress and immune responses in Nile tilapia, Oreochromis niloticus

PONE-D-24-21421R1

Dear Dr. Mustafa,

We’re pleased to inform you that your manuscript has been judged scientifically suitable for publication and will be formally accepted for publication once it meets all outstanding technical requirements.

Kind regards,

Ishtiyaq Ahmad, Ph.D

Academic Editor

PLOS ONE

Additional Editor Comments (optional):

Reviewers' comments:

Reviewer's Responses to Questions

**Comments to the Author**

1. If the authors have adequately addressed your comments raised in a previous round of review and you feel that this manuscript is now acceptable for publication, you may indicate that here to bypass the “Comments to the Author” section, enter your conflict of interest statement in the “Confidential to Editor” section, and submit your "Accept" recommendation.

Reviewer #1: All comments have been addressed

2. Is the manuscript technically sound, and do the data support the conclusions?

Reviewer #1: Yes

3. Has the statistical analysis been performed appropriately and rigorously? 

Reviewer #1: Yes

4. Have the authors made all data underlying the findings in their manuscript fully available?

Reviewer #1: Yes

5. Is the manuscript presented in an intelligible fashion and written in standard English?

Reviewer #1: Yes

6. Review Comments to the Author

Reviewer #1: Manuscript Number: PONE-D-24-21421R1

Title: Role of Phyllanthus niruri on the modulation of stress and immune responses in Nile tilapia, Oreochromis niloticus

Journal: PLOS ONE

The revised Ms is much improved based on the comments provided by the reviewers. Now, I can suggest this revised Ms for consideration after few typho errors rectified as below:

Line 40: “[2,3,4]” change to “[2-4]”

Line 52: include ‘were’ after Changes

Line 53: inclide ‘was’ after For example, fish

Line 120: Any clinical health checkup in the acclimation period?

Line 183, 185: Na2HPO4, of this 2 and 4 is change to be subscript

Line 209: andHDCF, need space between “and HDCF”

7. PLOS authors have the option to publish the peer review history of their article (what does this mean?). If published, this will include your full peer review and any attached files.

Reviewer #1: No

---

## [Editor Report · Acceptance letter]

26 Aug 2024

PONE-D-24-21421R1 

PLOS ONE

Dear Dr. Mustafa, 

I'm pleased to inform you that your manuscript has been deemed suitable for publication in PLOS ONE. Congratulations! Your manuscript is now being handed over to our production team.

Kind regards, 

on behalf of

Dr. Ishtiyaq Ahmad 

Academic Editor

PLOS ONE